# Comparison of 30-Day MACE between Immediate versus Staged Complete Revascularization in Acute Myocardial Infarction with Multivessel Disease, and the Effect of Coronary Lesion Complexity

**DOI:** 10.3390/medicina55020051

**Published:** 2019-02-15

**Authors:** Cem Doğan, Zübeyde Bayram, Murat Çap, Flora Özkalaycı, Tuba Unkun, Emrah Erdoğan, Abdulkadir Uslu, Rezzan Deniz Acar, Busra Guvendi, Özgur Yaşar Akbal, Ali Karagöz, Aykun Hakgor, Ahmet Karaduman, Samet Uysal, Ahmet Aykan, Cihangir Kaymaz, Nihal Özdemir

**Affiliations:** 1Department of Cardiology, Kosuyolu Heart Education and Research Hospital, University of Health Sciences, 34844 Istanbul, Turkey; zbydbyrm@hotmail.com (Z.B.); murat00418@hotmail.com (M.Ç.); tuba_unku3921@hotmail.com (T.U.); dremraherdogan49@gmail.com (E.E.); usludr83@hotmail.com (A.U.); denizacar_1999@yahoo.com (R.D.A.); guvendi2861@hotmail.com (B.G.); ozgurakball@hotmail.com (Ö.Y.A.); alikaragoz@yahoo.com (A.K.); aykunhakgor@hotmail.com (A.H.); ahmetkaraduman91@gmail.com (A.K.); sametuysal16@gmail.com (S.U.); cihangirkaymaz2002@yahoo.com (C.K.); ozdemirnihal@hotmail.com (N.Ö.); 2Department of Cardiology, Hisar Intercontinental Hospital, 34844 Istanbul, Turkey; florataniel@yahoo.com; 3Department of Cardiology, Kahraman Maraş Sütçü İmam University, 46000 Kahraman Maraş, Turkey; ahmetaykan@yahoo.com

**Keywords:** acute coronary syndrome, multivessel coronary artery disease, Syntax score

## Abstract

*Background and objective:* In patients with acute myocardial infarction and multivessel disease, the timing of intervention to non-culprit lesions is still a matter of debate, especially in patients without shock. This study aimed to compare the effect of multivessel intervention, performed at index percutaneous coronary intervention (PCI) (MVI-I) or index hospitalization (MVI-S), on the 30-day results of acute myocardial infarction (AMI), and to investigate the effect of coronary lesion complexity assessed by the Syntax (Sx) score on the timing of multivessel intervention. *Materials and methods:* We enrolled 180 patients with MVI-I, and 425 patients with MVI-S. The major adverse cardiovascular events (MACE) for this study were identified as mortality, nonfatal myocardial infarction, nonfatal stroke, acute heart failure, ischemia driven revascularization, major bleeding, and acute renal failure developed within 30 days. *Results:* The unadjusted MACE rates at 30 days were 11.2% and 5% among those who underwent MVI-I and MVI-S, respectively (OR 3.02; 95% confidence interval (CI) 1.51–6.02; *p* = 0.002). Associations were statistically significant after adjusting for covariates in the penalized multivariable model (adjusted OR 2.06; 95%CI 1.02–4.18; *p* = 0.043), propensity score adjusted multivariable model (adjusted OR 2.46; 95%CI 1.19–5.07; *p* = 0.015), and IPW (adjusted OR 2.11; 95%CI 1.28–3.47; *p* = 0.041). We found that the Syntax score of lesions did not affect the results. Conclusion: MVI-S was associated with a lower incidence of major adverse cardiovascular events within 30 days after discharge.

## 1. Introduction

Recent cardiovascular guidelines for the management of acute myocardial infarction (AMI) recommend that each patient should be treated with percutaneous coronary intervention (PCI) if he/she is amenable to an invasive approach. In the catheterization laboratory, multivessel disease is detected in approximately 40% of patients with ST segment elevation myocardial infarction (STEMI), and 70% of patients with non-ST segment elevation myocardial infarction (NSTEMI) [1,2]. A close association between multivessel disease and worse outcomes is well established [3,4,5], and recent randomized trials have verified the clear benefit of multivessel interventions for these patients [6,7,8,9]. In light of these studies, guidelines have changed in the direction of multivessel intervention [10,11,12]. Although certain strategies have been developed, the exact timing of complete revascularization (multivessel PCI) in AMI (index PCI or staged) remains a matter of debate.

In multivessel disease, it is obvious that the prognosis is related to the features (location, complexity, etc.) of the lesions, rather than the number of diseased vessels. A syntax score was developed that could predict prognosis. It has been proven to indicate the nature of the lesions, and it is also valid for estimating the prognosis of AMI patients [13,14,15].

In this study, we aimed to compare the efficacy of successful multivessel intervention on index PCI and index hospitalization, measured by 30-day outcomes, in AMI patients. In addition, we investigated the impact of coronary lesion complexity, assessed by the Sx score, on the short term clinical outcomes of these two PCI strategies.

## 2. Materials and Methods

In this observational study, we examined the effect of multivessel intervention during index PCI (MVI-I) versus index hospitalization (MVI-S) on post discharge 30-day mortality, and adverse cardiovascular events in patients with AMI, excluding those in cardiogenic shock. A Sx score was calculated for each subject. Patients who had previously undergone coronary artery bypass graft surgery (CABG) or PCI were not included, as the efficacy of the Sx score has been validated only for native vessels. Additionally, patients with left main coronary lesions (>50% stenosis in the left main coronary artery (LMCA) were excluded, because CABG might be a more suitable option in these cases. Data on the medical history, and the clinical and demographic data of the patients were obtained from the digital data bases of the institution. The first reference values registered in the system were recorded as laboratory values. Finally, a left ventricular ejection fraction (LVEF) measured during the pre-discharge echocardiographic evaluation was noted.

### 2.1. Study Population

This observational study included 2128 patients with AMI, who were admitted to our hospital between 2014 and 2017 and were treated with PCI. Overall, 191 patients with previous PCI, 95 with previous CABG, 69 in cardiogenic shock, 81 with significant LMCA lesions (>50% stenosis), and 818 with single vessel disease were excluded from the study. A total of 14 patients out of these single vessel intervention patients failed PCI story, and no staged PCI was performed for the other single vessel intervention patients during the hospitalization period or after 30 days of follow-up. The patients on whom PCI could not be performed for culprit or non-culprit coronary artery lesions (diameter<2 mm, diffuse disease, chronic total occlusion, etc.) or who had been revascularized during previous hospitalizations, and those patients whose Sx score could not be calculated, were excluded. The incidence of in-hospital major adverse cardiovascular events (MACE)in patients with multi-vessel disease was 11. There were four in-hospital deaths in the multivessel disease patients. These patients were also excluded from the study. Although both the staged PCI during index hospitalization (in the acute phase of MI) and after index hospitalization (in the subacute to chronic phase of MI) are still acceptable if patients were hemodynamically stable and have no severe residual ischemia; in our hospital, we prefer staged PCI at index hospitalization as a standard procedure. Our study population comprised of 180 MVI-I and 425 MVI-S patients, whose culprit and no culprit lesions were identified as eligible for PCI by the operator (vessel diameter>2 mm, ≥70% stenosis, not considering chronic total occlusion). Figure 1 shows a flowchart of the study. A local ethics committee approved the study (Saglik Bilimleri University Karta lKosuyolu Yuksek Ihtisas Egitimve Arastırma Hastanesi Ethics committee, approval date: 20 December 2018; issue number: 2018.9/5-144). This study conforms to the principles of the Helsinki declaration.

### 2.2. Definitions

The criteria for the diagnosis of AMI were as follows: Patients with chest pain that persisted for at least 30 min, new ST elevation at the J-point in two contiguous leads with cut points of ≥0.1mV in all of the leads (with the exception of leads V2–V3, where the following cut points applied:≥0.2 mV in men ≥40 years, ≥0.25 mV in men aged <40 years, and≥0.15 mV in women), new horizontal or down sloping ST depression ≥0.05 mV in two contiguous leads and/or a T inversion of ≥0.1 mV in two contiguous leads with a prominent R wave or R/S ratio of >1, new-onset left bundle branch block on a 12-lead electrocardiogram, and elevated cardiac markers such as creatine kinase-myocardial band or troponin I. The PCI procedures were performed by invasive cardiologists who are experienced in this field (>75 primary PCI/year).

The major adverse cardiovascular events (MACE) for this study were as follows: cardiovascular (CV) mortality, nonfatal myocardial infarction, nonfatal stroke, acute heart failure, ischemia driven revascularization (IDR), major bleeding, and acute renal failure developed within 30 days after discharge. The patient social security reports, hospital admission data, and death records for 30 days after discharge were examined in detail. Inadequate and incompatible information was verified by phone calls.

In this study, acute heart failure refers to the rapid onset or worsening of symptoms and/or signs of heart failure (includes denovo heart failure, and the decompensation of previous heart failure or cardiogenic shock, which requires advanced treatment). Major bleeding was evaluated according to the thrombolysis in myocardial infarction (TIMI) bleeding classification. Repeated PCI or CABG of both culprit and non-culprit vessels, driven by ischemia (pain, ischemic ECG changes, or other symptoms and signs presumed to be of ischemic origin), were designated as IDR. Acute renal failure, including contrast induced nephropathy, was defined using the creatinine criteria and the ‘’Acute Kidney Injury definition of the Risk, Injury, Failure, Loss, End-stage kidney disease’’ (RIFLE) consensus [16].

### 2.3. Sx Score

The Sx score was calculated using software available at http://www.sytaxscore.com. The NSTEMI scores were calculated from the first angiographic evaluation of the patients. In STEMI patients, the score was calculated from the initial angiographic image in the presence of a TIMI II–III flow in the culprit artery, whereas in the TIMI 0–I blood flow, the scoring was performed on the subsequent images after the distal TIMI II-–II flow obtained by wiring/balloon inflation. In previous STEMI trials, patients with TIMI 0–I currents had an average score of five points higher than the baseline scores (as the software accepted lesions as chronic total occlusion). The Sx score calculated after the wiring reflected the underlying anatomy more clearly in the culprit vessels, and the scores both before and after the wiring/balloon inflation are associated with the prognosis [17]. Therefore, we used the scoring after wiring/balloon inflation to provide homogeneity when calculating the scores in the NSTEMI and STEMI patients.

### 2.4. Statistical Analysis

Numerical variables were presented with a median and interquartile range, and the categorical data were presented with percent and *n*. The major adverse cardiovascular events (MACE) for this study were identified as mortality, nonfatal myocardial infarction, nonfatal stroke, acute heart failure, ischemia driven revascularization (IDR), major bleeding, and acute renal failure developed within 30 days after discharge. It is very important that the candidate predictors for MACE should be clinically and biologically plausible, and their relationships with short-term MACE should be demonstrated in previous studies. We determined the variables we included in the model according to these principles. We identified age, MVI timing (MVI-I or MVI-S), STEMI-NSTEMI, Syntax score, LVEF, and gender as candidate predictors. The variables with very low or very high frequencies were not included in the model. As a result, we included these six candidate variables in our final model.

In order to develop a clinical model, the sample should be large enough and the number of predictors should be sufficiently conservative. Specifically, there should be at least 10 patients with outcome (outcome/variable>10) compared to the number of candidate predictors taken into the model. In our study, MACE was present in 35 patients, while six candidates were identified in our model (35/6 = 6). Therefore, the penalized maximum likelihood estimation (PMLE) method was used instead of the traditional logistic regression, in order to reduce the risk of overfitting. Numerical variables such as age, Sx, and LVEF were included in the model as flexible smooth parameters using the restricted cubic spline. PMLE maximizes the penalized log likelihood, instead of maximizing the log likelihood made in the traditional logistic regression. Therefore, the maximum log likelihood of the model is adjusted by the penalty factor. The relative importance of each predictor in the models was estimated with a partial X2 value for each predictor, divided by the model’s total X2, which estimates the independent contribution of the predictor to the variance of the outcome. The calibration was assessed by plotting the observed outcome on the y-axis and the predicted outcome in the x-axis. Deviations from the 45-degree line will show the bias for the outcome. The Loess algorithm was used for the relationship between the observed outcome and predicted outcome. The propensity scores (PS) were calculated using a multivariable logistic regression model with the dependent outcome as treatment with MVI-S vs. MVI-I. In order to evaluate the associations between the treatment group (MVI-S vs. MVI-I) and the primary outcome, we used the spline function of the logit propensity score. We also used propensity scores with inverse probability weighting (IPW) in order to adjust for the differences between the two treatment groups.

All of the statistical analyzes were performed using R-software v. 3.5.1 (R statistical software, Institute for Statistics and Mathematics, Vienna, Austria).

## 3. Results

The median age of the 605 patients with AMI and multivessel disease was 58 (50.5–66) years; 487 patients (80.5%) were males. Overall, 351 patients (58%) were admitted with STEMI, and 254 patients (42%) with NSTEMI. The mean Sx score of all of the patients was 15.2 ± 5.8. In total, 180 patients (29.8%) underwent multivessel intervention during index PCI (MVI-I), and 425 patients (70.2%) underwent staged multivessel intervention at the index hospitalization (MVI-S). The MVI-I and MVI-S rates in the STEMI patients were 26% vs. 74%, and 35% vs. 65% in the NSTEMI patients. The median time between the first and second PCI was 48 (22–63) hours in the MVI-S group. The baseline demographic, procedural characteristics, and medications of the MVI-S and MVI-I groups are shown in Table 1.

The median Sx scores of the MVI-S and MVI-I groups were not different. In addition, when we evaluated the groups as low (0–22), moderate (23–32), and high (>32) risk, according to Sx risk categories, no difference was observed.

The unadjusted MACE rates at 30 days were 11.2% and 5% among those who underwent MVI-I and MVI-S, respectively (unadjusted OR = 3.02; 95% CI 1.51–6.02; *p* = 0.002). This association was statistically significant after adjusting for covariates in the penalized multivariable model (adjusted OR = 2.06; 95%CI 1.02–4.18; *p* = 0.043). Also, associations between the MVI timing and MACE remained significant using both the propensity score adjusted multivariable model (adjusted OR = 2.46; 95%CI 1.19–5.07; *p* = 0.015) and the IPW (adjusted OR = 2.11; 95%CI 1.28–3.47; *p* = 0.041) (Table 2). MVI timing and LVEF were ranked as the strongest predictors of MACE, contributing 80% of the explainable outcome variation in our model (Figure 2). The relationship between the log odds of the MACE and regression modelling of all of the variables are shown in Figure 3. There was no significant interaction between the MI type (STEMI/NSTEMI) and intervention timing (MVI-S/MVI-I) among those patients enrolled to the study (*p* for interaction 0.534, Figure 4). There was a moderate agreement on the calibration plot, which demonstrates the relationship between the observed and predicted outcome. Our model slightly overestimated the observed risk in case of the predicted risk being 0.3 and above (Figure 5).

## 4. Discussion

In this study, we examined the association between multivessel intervention timing (MVI-I vs. MVI-S) and short-term clinical outcomes. The rates of MVI-I and MVI-S were similar between the STEMI (26% vs. 74%) and NSTEMI (35% vs. 65%) patients. Consistent with previous studies, we demonstrated that the incidences of MACEs were significantly less in the patients with AMI receiving staged multivessel intervention at index hospitalization, than the patients receiving multivessel intervention at index procedures. In addition, we could not find a significant interaction between the coronary lesion complexity, assessed by the Sx score, and the clinical outcomes of both MVI timings.

Within the clinic parameters used in the Sx score II, we took the age, gender, and LVEF into our model. We found no statistically significant relationship between age and gender with MACE, but we found LVEF to be a significant predictor of MACE.

Multivessel disease is common in AMI (both STEMI and NSTEMI), and is associated with poorer outcomes [3,4,5]. According to the pooled data from the eight randomized, controlled trials of STEMI, an increase in mortality was observed at 30 days if additional non-culprit stenotic lesions existed [18].

There are various studies in the literature indicating that either index or staged multivessel revascularization is beneficial for the treatment of patients with acute coronary syndrome. Two randomized trials compared only the culprit vessel intervention with PCI of all vessels in acute coronary syndrome. In the Preventive Angioplasty in Acute Myocardial Infarction (PRAMI) trial, “preventive” PCI of a non-infarct-related artery was associated with a reduced risk of the composite end point of death, AMI, or refractory angina, compared with IRA-only PCI [6]. In the CvLPRIT trial, a significant reduction in the primary end point of death, recurrent AMI, heart failure, or IDR after12 months was found with MV-PCI [7]. The effectiveness of the multivessel intervention was irrespective of the timing of the non-culprit lesion revascularization in these studies.

The DANAMI-3 PRIMULTI and Primary Angioplasty in Patients Transferred from General Community Hospitals to Specialized PTCA Units with or without Emergency Thrombolysis (PRAGUE-13) trials evaluated staged revascularization in multivessel disease. In the DANAMI-3 PRIMULTI trial, the composite primary outcome of the all-cause death, nonfatal AMI, or IDR of non-IRA lesions occurred less often the inpatients assigned to staged PCI before discharge [8]. The PRAGUE-13 trial failed to show a difference between the two strategies in the composite primary end point of all-cause death, nonfatal AMI, and stroke [9]. Apart from these, the fractional flow reserve-guided complete revascularization reduced the MACE rates in the Compare-Acute trial [19].

In our study, MACEs at 30 days were nearly two times higher in the MVI-I than MVI-S. The lower incidences of MACEs in the MVI-S remained significant in both a propensity score adjusted multivariable model (adjusted OR 2.46; 95%CI 1.19–5.07; *p* = 0.015) and IPW (adjusted OR 2.11; 95%CI 1.28–3.47; *p* = 0.041). When we look at the reasons for the differences in our results from previous studies, there is heterogeneity in the inclusion and exclusion criteria, PCI timing and techniques, end points, and methods of analysis of them. Moreover, the study populations are not sufficiently large, and the relatively low frequency of the hard endpoints, such as death and AMI (only PRAMI deaths and AMI decrease), does not allow for a precise decision to be reached about the superiority of one method over the other, a situation that reflects the real world. Furthermore, the subgroup of patients with complex lesions (ostial and bifurcation) who would benefit the most from revascularization was not recruited in these studies. In these randomized studies, PRAMI was the only trial that specifically looked at non-CVI at the time of primary PCI, whereas the remaining trials examined non-CVI as a staged procedure during the index in-hospital admission; the timing of non-culprit vessel revascularization remained equivocal.

In light of these trials, the ESC 2017 guideline for the management of STEMI recommends that the routine revascularization of non-culprit lesions should be considered before hospital discharge [10]. The 2015 ACC/AHA/SCAI focused update on primary PCI for patients with STEMI recommends that PCI to a non-IRA may be considered in selected patients who have STEMI and multivessel disease, with a desired stability at the time of both primary PCI and the elective staged procedure [12].

Evidence for the effectiveness of multivessel intervention in NSTEMI patients is scant. Several studies have shown that multivessel intervention is a safe and effective treatment strategy for NSTEMI [20,21]. In deciding whether to perform multivessel intervention in this patient group, it is recommended to determine not only the clinical status and comorbidities of the patient, but also the complexity and distribution of the lesions (including the calculation of the Sx score) [11].

When we evaluated the pathophysiological causes of a low rate of MACE in the MVI-S group, we could look at the potential disadvantages of MVI-I, which includes increased contrast and radiation burden, increased periprocedural MI due to prothrombotic and inflammatory environment in MI, overestimation of lesions due to spasm, prolongation of procedure, and interventions for the treatment of additional lesions resulting in hemodynamic and mechanically increased risk of instability [22,23,24]. Furthermore, in the present study, the incidences of certain poor prognostic factors, such as anterior MI, left anterior descending artery as the culprit artery, and Killip class two, were higher in the MVI-I group [25,26,27].

There are registries and meta-analyses in the literature that support our results. In Iqbal’s registry, MVI-S was associated with lower mortality rates than either MVI-I or CO-PCI, and MVI-S reduced the need for repeated revascularization in comparison with CO-PCI [28]. In the meta-analysis of Vlaar, which compared MVI-I, MVI-S, and CO-PCI, it was stated that the short-term and long-term mortality rates were lower in the patients receiving MVI-S [29]. In another meta-analysis, Li and colleagues compared MVI-I and MVI-S. MVI-S was found to have lower in-hospital and long-term mortality rates, with favorable safety parameters [30]. In the meta-analysis by Barney et al., staged PCI had short- and long-term survival benefits, and decreased the requirements for repeated PCI compared with CO-PCI [31]. In another meta-analysis, the mortality of MVI-S was lower than that of both MVI-I and CO-PCI [32]. When patients with NSTEMI and STEMI were compared, MVI-S was associated with lower short-term MACE rates in patients with STEMI, although the *p*-value did not reach statistical significance. The results of two new prospective studies on the timing of multivessel intervention, COMPLETE (NCT01740479) and FULL REVASC (NCT02862119), may show us more clearly.

In our analysis, while the MVI timing and LVEF were found to be strong predictors for MACE at 30 days, there was no relation between the Sx score and incidence of MACE. Previous studies have shown that the Sx score was>22 in high-risk patients [33]. In our study, patients with acute coronary syndrome who were relatively stable were included in the study. The median Sx scores were found to be 14 (low risk) in both groups because of exclusion criteria such as the significant LMCA lesion, chronic total occlusion, and previous CABG. Both the MVI-S and MVI-I patients had aSx score <22 in 89% of the patients. The reason for not finding the interaction between the MACE and Sx score can be due to the fact that both groups are comparatively low risk.

### Limitations

This study was a single-center, retrospective study with potential selection biases. The decision to perform a multivessel intervention at the index PCI or at the index hospitalization was at the discretion of the operator. Our study population was small and heterogeneous, including both STEMI and NSTEMI. Patients who had failed PCI and experienced in-hospital MACE were not included in our study. A longer follow-up period will be required for a more complete evaluation of the effects of MVI-S and MVI-I on MACE, especially on mortality.

## 5. Conclusions

In this observational trial, we found that MVI-S was associated with a lower incidence of MACE within 30 days after discharge in patients with AMI with multivessel disease. We could not find a significant relationship between the MACE and Sx score. This issue may be clarified with further prospective randomized studies in those patients presented with AMI in the absence of cardiogenic shock and those diagnosed with multivessel CAD.

## Figures and Tables

**Figure 1 medicina-55-00051-f001:**
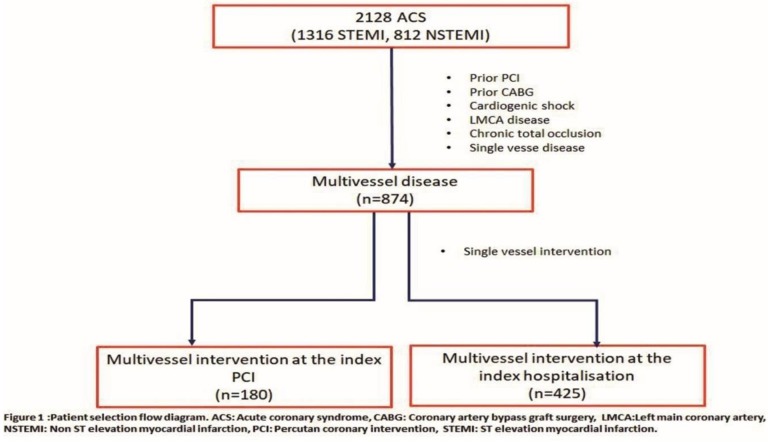
Flow chart of the study.

**Figure 2 medicina-55-00051-f002:**
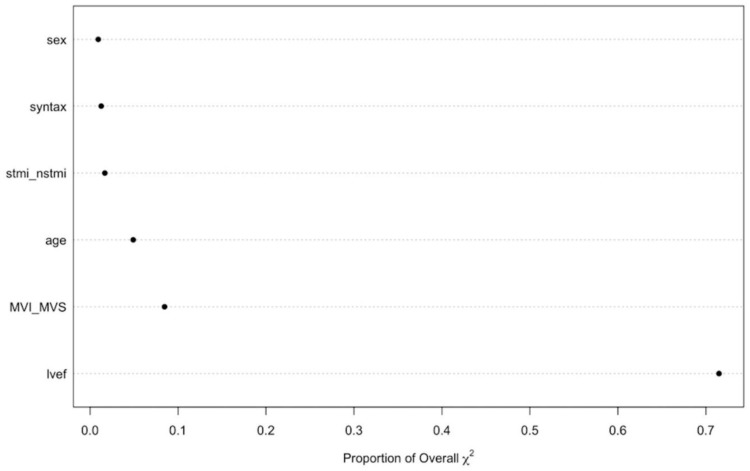
Importance of individual predictors. The importance of each predictor in our model was calculated as the proportion of the explainable outcome variation contributed by each predictor (partial χ^2^ value for each predictor divided by the model’s total χ^2^).

**Figure 3 medicina-55-00051-f003:**
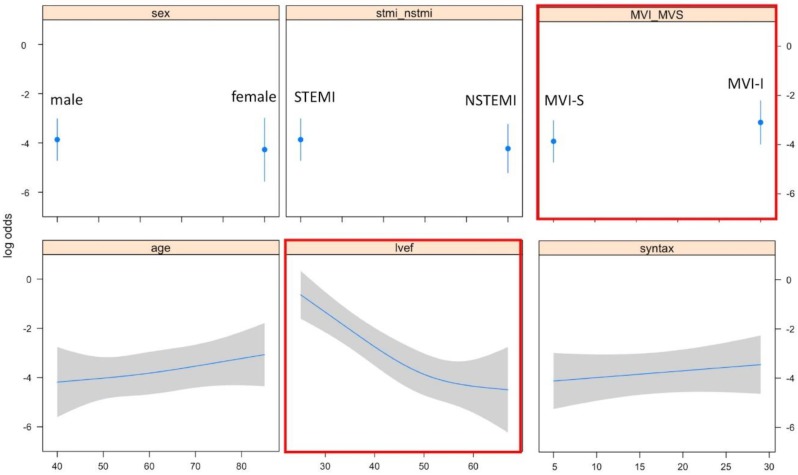
Calibration curve illustrates the calibration accuracy of the original model (apparent) and the bootstrap model (bias corrected) for 30-day MACE with locally weighted scatterplot smoothing used to model the relation between the actual and predicted probabilities.

**Figure 4 medicina-55-00051-f004:**
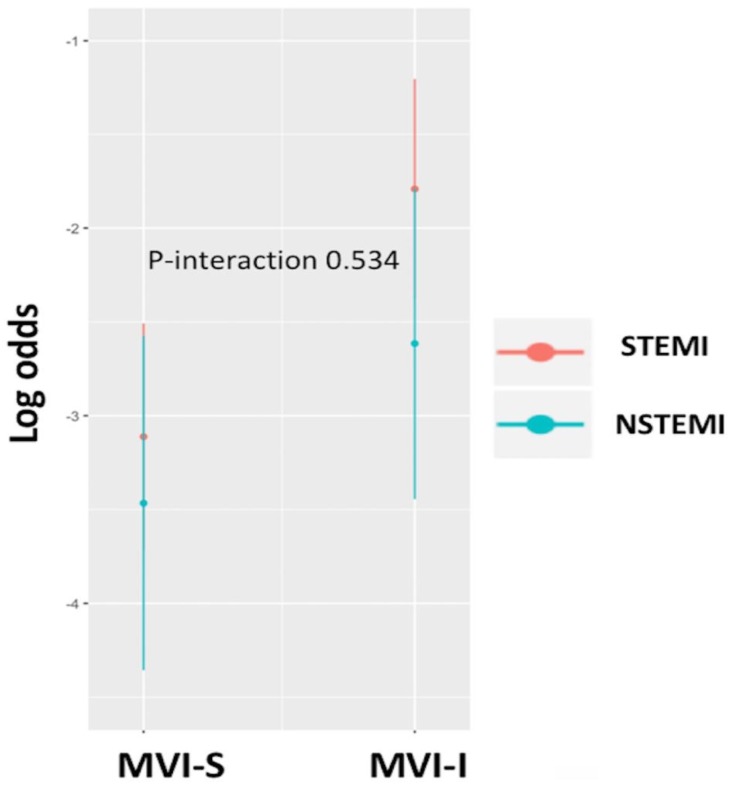
Interaction between MI type (STEMI/NSTEMI) and MVI timing (MVI-S/MVI-I).

**Figure 5 medicina-55-00051-f005:**
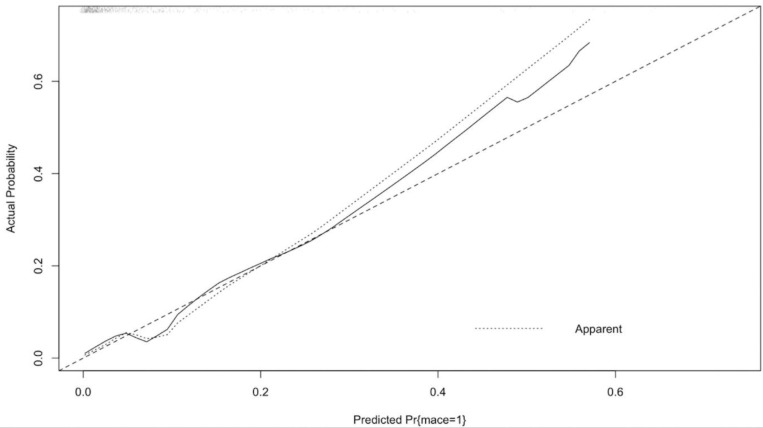
Calibration plot showing relation between the observed and predicted outcome.

**Table 1 medicina-55-00051-t001:** Baseline demographic, procedural characteristics, and medications.

Variables	MVI-S (n = 425)	MVI-I (n = 180)	*p* Value
Age, median (interquartile range (IQR)), years	58 (51–66)	59 (50–67)	0.815
Male, n (%)	335 (78.8)	152 (84.4)	0.111
Diabetes mellitus, n (%)	108 (25.4)	49 (27.2)	0.642
Hypertension, n (%)	334 (78.6)	146 (81)	0.483
Smoking, n (%)	184 (43.3)	84 (46.7)	0.615
Hyperlipidemia n (%)	269 (63.3)	120 (66.7)	0.429
Ejection fraction, median (IQR), %	50 (45–60)	50 (40–55)	0.333
Creatinine, median (IQR), mg/dL	0.8 (0.7–1.0)	0.9 (0.7–1.0)	0.024
STEMI, n (%)	260 (61.2)	91 (50.6)	0.016
Non-STEMI, n (%)	165 (38.8)	89 (49.4)	
**STEMI type, n (%)**			0.044
Anterior	99 (37.9)	48 (52.7)	
Inferior	136 (52.1)	35 (38.5)	
Others	26 (10.0)	8 (8.8)	
**Culprit Lesion**, n (%)			0.016
LAD	153 (36.0)	86 (47.8)	
RCA	162 (38.1)	46 (25.6)	
CX	91 (21.4)	40 (22.2)	
Other	19 (4.5)	8 (4.4)	
**Syntax Score**, median (IQR)	14 (12-19)	14 (11–28)	0.623
0–22, %	85.4	89.4	0.001
23–32, %	13.6	9.4	
>32, %	0.9	1.1	
**KILLIP, n (%)**			
1	420 (98.8)	169 (93.9)	
2	5 (1.1)	11 (6.1)	
**Procedural characteristics**			
CV DES, n (%)	393 (92.2)	165 (91.7)	0.766
CV TIMI III, n (%)	366 (86.1)	169 (93.8)	0.891
CV GPIIb-IIIa inhibitor, n (%)	44 (11.7)	15 (8.4)	0.231
CV Stent diameter, median (IQR)	2.75 (2.5–3.0)	2.75 (2.5–3.0)	0.559
Thrombectomy, n (%)	33 (8.8)	9 (5.1)	0.123
**Medications, n (%)**			
ASA	425 (100)	180 (100)	1.000
P_2_Y_12_ inhibitor	422 (99.8)	179 (99.4)	0.822
Beta blocker	412 (98.3)	176 (98.9)	0.616
ACEI	399 (95.2)	164 (92.1)	0.136
Statin	408 (97.4)	177 (99.4)	0.100

Continuous data were compared using the Mann–Whitney U test (two-tailed) and the categorical, chi-square test (two-tailed). ACEI—angiotensin-converting enzyme inhibitor; ASA—acetyl salicylic acid; CV—culprit vessel; DES—drug eluting stent; CX—circumflex artery; HDL—high density lipoprotein; LAD—left anterior descending artery; LDL—low density lipoprotein; MVI—multivessel intervention; non-STEMI—myocardial infarction without ST segment elevation; RCA—right coronary artery; STEMI—myocardial infarction with ST segment elevation.

**Table 2 medicina-55-00051-t002:** Unadjusted and adjusted clinical outcomes at 30 days.

			Odds Ratios (95% Confidence Interval)			
	MVI-I n (%)	MVI-S n (%)	Unadjusted	PS Adjusted MV Model	IPW	Covariate Adjusted
**MACE**	20(11.2)	21 (5.0)	3.02 (1.516.02)	2.46 (1.19–5.07)	2.11 (1.28–3.47)	2.06 (1.02–4.18)
30 Day CV mortality	16 (8.9)	9 (2.1)	4.51 (1.96–10.4)			
Nonfatal MI	3 (1.7)	7 (1.7)	1.01 (0.26–3.96)			
Ischemia driven revascularization	3 (1.7)	10 (2.4)	0.70 (0.19–2.59)			
Acute heart failure	16 (8.9)	15 (3.5)	2.67 (1.29–5.52)			
Nonfatal Stroke	2 (1.1)	1 (0.2)	2.37 (0.15–38.1)			
Acute renal failure	9 (5.0)	15 (3.5)	1.44 (0.62–3.35)			
Bleeding	4 (2.2)	11 (2.6)	0.85 (0.27–2.72)			

CV—cardiovascular; IPW—inverse probability-weighting; MI—myocardial infarction; MV—multivariable; MVI-I—multivessel intervention at index PCI; MVI-S—multivessel intervention at index hospitalization; PS—propensity score.

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
