# Peer review of "Comparison of 30-Day MACE between Immediate versus Staged Complete Revascularization in Acute Myocardial Infarction with Multivessel Disease, and the Effect of Coronary Lesion Complexity"

_1010-660X, 2019, doi:10.3390/medicina55020051_

Round 1

Reviewer 1 Report

The study reports on 30 day MACE outcomes in patients with acute myocardial infarctions and multivessel disease, depending on wether they were treated for the non-culprit stenoses at the time of hospital presentation or later on during hospitalization. To understand why one of the approaches was chosen above the other or to understand any interaction with lesion severity and the outcomes between the two approaches, the angiographic appearance of the stenoses was studied using the SYNTAX score. 

The result was that the outcome was worse in patients treated with immediate complete revascularisation, and better in patients treated in a staged fashion. The SYNTAX score was low in both approaches, and the same, so that this did not explain the difference in outcome. After adjustment for unknown covariates, the results still pointed to a different outcome associated with the approach (immediate or staged procedure) that was chosen for the treatment. In a logistic model, the outcomes were mainly explained by LV function, but also by the interventional approach.

Comments

Figure 1 shows there were 2128 patients presented, and after exclusion of patients with single vessel disease (828) and other problems, 874 patients remain with multivessel disease. However, not all of these patients will be treated for multivessel disease, a total of 605 patients remain for analysis (the text lin 74 and figure 1 mention: "the trial", but it is not a trial). The question here is who are these patients that now remain to be compared to each other, did they all have a succesfull PCI in both or more vessels, while those who did not have a successfull PCI or those that did not have a further PCI were not included for study ? Any intent to repeat your study would fail, because there will be unsuccessful PCI's.  We do not know from the results in Table 1, it seems that all procedures were succesfull. What we do know, is that the median SYNTAX score of the patients was low. You should include all patients who were intended to have a multivessel PCI. 

In the Methods, it is not described when the follow up starts. It should start at the first day after the PCI for both patient groups. In this way, any mortality or morbidity of the first days after PCI will be a "reason" not to perform a second, staged PCI, and therefore there should have been patients who will not have had a staged PCI although they might have been planned for it. Or, if this was not done in your study, any patient who died before the staged PCI was done, was not included in your study and could have made a difference. Also, only when the follow up starts at day 1 of the first procedure, the question: "is it safe to wait for the second PCI" can be answered, because the MACE between the two procedures in the staged approach can only then be reported, something that the authors may not have thought about.   

In the Methods, no explanation is given on the analysis that is performed, with unknown co-variates, although the analysis itself was nicely put into a logistic model with predictors, but whether these were covariates? Interaction was tested between the NSTEMI/ STEMI group and the type of intervention (staged or immediate) for outcome, which is good. There may have been an interaction between the interventional approach and the LV function or the way the patient presented in higher Killip class.

It is not known what the time period was between first and second PCI procedure in the staged group. Was it very diverse ? Could it have made a difference ?

When the methods are not very clear, there is no conclusion possible.    

Author Response

Dear Rewiever

You've made us happy by evaluating our work. Thanks to your valuable contributions, we have re-submitted the manuscript that we have recreated by correcting the deficiencies we have seen, according to your suggestions. For us, your  your contributions to our new manuscript are very valuable. Also I tried to answer some of your questions.

Your sincerly

I apologize , When editing the manuscript, I noticed that the statistics part was not installed by mistake. I've added a detailed statistic section to the new manuscript.

The trial word in line 74 and figure 1 was changed to study.

In our study, we did not include patients who had a single vessel intervention ( as we mentioned in methodology patients on whom PCI could not be performed for culprit or non-culprit coronary artery lesions ,diameter< 2mm, diffuse disease, chronic total occlusion, etc., or who had been revascularized during previous hospitalizations were excluded) in 874 multi-vessel patients. Only 14 out of these single vessel intervention patients have failed PCI story and no staged PCI has been performed for the other single vessel intervention patients during the hospitalization period or 30 days of follow-up. Our study was a retrospective observational study and was not randomized. For this reason, we have tried to get the patients who are in accordance with the study protocol that we determined earlier. Patients with unsuccessful interventions or single-vessel intervention may have poor outcomes due to incomplete revascularization. Since our aim was to investigate the effect of multivessel interventions at different times on the outcome, we did not take the patients who were treated with single vessel intervention for the above mentioned reasons.

As mentioned in the methodology section, we have evaluated the MACEs after discharge. We did not evaluate the in-hospital  events. The incidence of in-hospital MACE in patients with multi-vessel disease was 11. There  were 4 in-hospital death in multivessel disease patients. In our retrospective study, we do not know which of these patients was intended to be staged PCI and which one was intended to be abandoned by singel vessel intervention.

In the statistical analysis section we added, the analysis methods applied, why these methods were selected, which parameters were taken as covariate was specified. In our study, high-risk patients, such as shock patients, were excluded, and both groups included Killip I and II, which were slightly more stable. Interaction between intervention type (index-staged) and LV function and Killip was examined but no statistically significant interaction was found.

The median time between first and second PCI was 48 (22-63) hours in MVI-S group (we added in results section of new manuscript).

Reviewer 2 Report

The manuscript entitled "Comparison of Short Term Results of Complete Revascularization at Index Percutaneous Coronary Intervention Versus Staged Revascularization in Acute Myocardial Infarction, and the Effect of Coronary Lesion Complexity on These Results" is an observational study on 605 patients who had multivessel intervention during index PCI (MVI-I) versus index hospitalization (MVI-S) and compared 30-day mortality and adverse cardiovascular events in patients with AMI of both groups. It is a very well written article that covers all the aspects of both interventions. 

The title is a  good reflection of the whole article.

The abstract is a relevant, brief overview of the whole article which summarized the main findings.

All the appropriate studies were referenced.

Outcomes are very well chosen.

Results and charts are comprehensive.

All the current guidelines are thoroughly discussed.

Overall, it is an excellent article with good review of the topic.

Author Response

Dear Reviewer

First of all, thank you for reviewing our article. Your valuable comments excited us. We made some changes in the Manuscript in line with the criticism of other reviwer, we re-submitted.

Reviewer 3 Report

In this observational, retrospective, non-randomized single-centre study the authors tried to investigate the effect of multivessel PCI timing on 30-day outcomes of AMI and evaluate interaction between anatomical Syntax score and the timing of multivessel PCI.  

Major comments:

1) Despite the study was aimed to investigate the effect of coronary lesion complexity assessed by Syntax score on the tining of multivessel intervention, the authors did not elaborated sufficiently on this issue in the Results, Discussion and Conclusion sections. The interaction between Syntax score and MACE did not assessed. 

Minor comments:

1) The distribution of patients in according to the Sx risk categories (low: 0-22, intermediate 23-32; high: >32) should be added in Table 1.

2) Row 118: "Baseline demographic and procedural characteristics and medications of overall, MVI-S and MVI-I groups was shown in Table 1". There is no column for overall study group in Table 1. 

3) Abbreviations IPW, IDR and FFR should be expanded in the text of the manuscript.

4) Figure 4 is excessive.

5) The number of PCIs performed by one operator per year in the study centre should be indicated in the Material and Methods.

6) As compared to the anatomical SYNTAX score, SYNTAX score II include clinical predictors (age, female sex, LVEF etc). The present study did not reveal interaction between 30-day outcomes and common mortality predictors (age, for example). This issue should be discussed briefly.

7) Age is an important clinical factor for patients with PCI (see DOIs: 10.1093/eurheartj/ehi187, 10.1016/j.crvasa.2014.03.004, etc.). But in the presented study, age did not manifest itself as a significant indicator. Discuss this fact in more detail in the Discussion.

Author Response

Dear Reviewer

You've made us happy by evaluating our work. Thanks to your valuable contributions, we have re-submitted the manuscript that we have recreated by correcting the deficiencies we have seen, according to your suggestions. For us, your  your contributions to our new manuscript are very valuable. Also I tried to answer some of your questions.

Your sciencerly

I apologize , When editing the manuscript, I noticed that the statistics part was not installed by mistake. I've added a detailed statistic section to the new manuscript.

In our study, we could not find a significant interaction between Sx score and MACE. We added our evaluations about Sx score to the results and discussion section according to your suggestions.

The distribution of patients in according to the Sx risk categories (low: 0-22, intermediate 23-32; high: >32)  added in Table 1.

The word overall in line 118 was written by mistakenly. We deleted it.

Abbreviations IPW, IDR and FFR  expanded in the text of the manuscript.

Figure 4 was not excluded from the manuscript in line with other rewivers.

The number of PCIs performed by one operator per year  indicated in the Material and Methods.

In line with your suggestions, we have added our findings and comments on clinical parameteres such as age, gender, and LVEF to the relevant manuscript sections.

WE added a more detailed evaluations of us about interaction between age and MACE in discussion.

Round 2

Reviewer 1 Report

The study, which has been resubmitted with Methods included, reports on 30 day MACE outcomes in patients with acute myocardial infarctions and multivessel disease, with the question whether outcomes may differ in patients by treating the non-culprit stenoses immediately during the first PCI or somewhat later during hospitalisation. It is a retrospective study. To understand why one of the approaches was chosen above the other or to understand any interaction with lesion severity and the outcomes between the two approaches, the angiographic appearance of the stenoses was studied using the SYNTAX score. Further, to understand if the method of immediate PCI or staged PCI was contributing to outcomes, a multivariate analysis was done on MACE with several clinical variables of known importance (age, LVEF).

The result was that the outcome was worse in patients treated with immediate complete revascularisation (11,2% MACE), and better in patients treated in a staged fashion (5% MACE). The SYNTAX score was low in both approaches, and the same, so that this did not explain the difference in outcome. After adjustment for unknown covariates, the results still pointed to a different outcome associated with the approach (immediate or staged procedure) that was chosen for the treatment. In a logistic model, the outcomes were mainly explained by LV function, but also by the interventional approach.

Comments (also with previous answers of authors).

METHODS section is now included with statistical plan, which is quite good. 

1.  Selection of patients. Figure 1 shows there were 2128 patients presented, and after exclusion of patients with single vessel disease (828) and other problems, 874 patients remain with multivessel disease. However, not all of these patients will be treated for multivessel disease, a total of 605 patients remain for analysis (the text lin 74 and figure 1 mention: "the trial", but it is not a trial). The question here is who are these patients that now remain to be compared to each other, did they all have a succesfull PCI in both or more vessels, while those who did not have a successfull PCI or those that did not have a further PCI were not included for study ?

Answer of authors: 

In our study, we did not include patients who had a single vessel intervention ( as we mentioned in methodology patients on whom PCI could not be performed for culprit or non-culprit coronary artery lesions ,diameter< 2mm, diffuse disease, chronic total occlusion, etc., or who had been revascularized during previous hospitalizations were excluded) in 874 multi-vessel patients. Only 14 out of these single vessel intervention patients have failed PCI story and no staged PCI has been performed for the other single vessel intervention patients during the hospitalization period or 30 days of follow-up. Our study was a retrospective observational study and was not randomized. For this reason, we have tried to get the patients who are in accordance with the study protocol that we determined earlier. Patients with unsuccessful interventions or single-vessel intervention may have poor outcomes due to incomplete revascularization. Since our aim was to investigate the effect of multivessel interventions at different times on the outcome, we did not take the patients who were treated with single vessel intervention for the above mentioned reasons.

My comment on this selection:

The authors should state in the Figure 1 and text, that 14 patients were intended for multivessel PCI (immediate or staged, please specify) but had a failed PCI. Also state that the other 269-14 = 255 pts with multivessel disease were not accepted/ suitable for multivessel PCI. This gives us an indication of real world experience, in which 255/874 pts = 29% are not eligible for multivessel PCI, although according to the guidelines, there would be an indication.   

By not selecting your failed PCI patients, you should be aware that conclusion on an intention to perform a immediate or staged is now flawed by only selecting the successful cases. This is a limitation that you will have to describe. 

2. MACE at 30 days definition. In the Methods, it is not described when the follow up starts. It should start at the first day after the PCI for both patient groups. In this way, any mortality or morbidity of the first days after PCI will be a "reason" not to perform a second, staged PCI, and therefore there should have been patients who will not have had a staged PCI although they might have been planned for it. Or, if this was not done in your study, any patient who died before the staged PCI was done, was not included in your study and could have made a difference. Also, only when the follow up starts at day 1 of the first procedure, the question: "is it safe to wait for the second PCI" can be answered, because the MACE between the two procedures in the staged approach can only then be reported, something that the authors may not have thought about.  

Answer of the authors:

As mentioned in the methodology section, we have evaluated the MACEs after discharge. We did not evaluate the in-hospital events. The incidence of in-hospital MACE in patients with multi-vessel disease was 11. There were 4 in-hospital death in multivessel disease patients. In our retrospective study, we do not know which of these patients was intended to be staged PCI and which one was intended to be abandoned by singel vessel intervention.

My comment on this definition:

The authors have now described in Methods that the MACE follow up starts at discharge from the hospital. This is obviously a limitation, because any in hospital deaths are not reported in MACE, while the outcome is seen to be a result from the PCI at admission. The authors should describe this very clearly in Methods: include post discharge 30-day.. (line 56). Also, in conclusion, it is now stated that "MVI-S was associated with a lower incidence of major adverse cardiovascular events within 30 days", which is not very clear to me. It should be: within 30 days after discharge. Please change line 301 and abstract line 32. In discussion on the MACE outcome, other studies, especially the randomized studies, will not have done this in this way, but will have reported outcomes starting from the day of the first PCI. 

The authors should report in Results: "The incidence of in-hospital MACE in patients with multi-vessel disease was 11. There were 4 in-hospital deaths in multivessel disease patients"; which can be inserted before the reporting of the MACE in line 173. I think that these 4 patients were one of the 874 patients that were not included in the final study, and therefore Figure 1 should be adapted not only with the 14 failed PCI patients who were excluded, but also with these 4 deaths.   

3. In the Methods, no explanation is given on the analysis that is performed, with unknown co-variates, although the analysis itself was nicely put into a logistic model with predictors, but whether these were covariates? Interaction was tested between the NSTEMI/ STEMI group and the type of intervention (staged or immediate) for outcome, which is good. There may have been an interaction between the interventional approach and the LV function or the way the patient presented in higher Killip class.

It is not known what the time period was between first and second PCI procedure in the staged group. Was it very diverse ? Could it have made a difference ?

Answer of the authors

In the statistical analysis section we added, the analysis methods applied, why these methods were selected, which parameters were taken as covariate was specified. In our study, high-risk patients, such as shock patients, were excluded, and both groups included Killip I and II, which were slightly more stable. Interaction between intervention type (index-staged) and LV function and Killip was examined but no statistically significant interaction was found.

The median time between first and second PCI was 48 (22-63) hours in MVI-S group (we added in results section of new manuscript).

My comment on the answer

Methods section has been adapted accordingly, thank you. Also in results is now visible what was the delay in staged procedures (line 159).

Line 158 can be adapted: instead of %26, %74, write 26% and 74%, als for the other two numbers. Apparently, there is some information in STEMI versus NSTEMI as to the choice of intervention timing, but the results are that there was no interaction (line 182-183 and fig 4), so this is adequately approached. Whether there was an interaction between the main predictor of events (= the LVEF) and the choice of timing for intervention (my original question) was not investigated further, but it doesn't have such interest since the model in figure 2 separates the influences of both. It would only be of interest to see why the operators made their choice to stage an intervention or not (ALL or NOTHING approach in patients that are severly ill, or the other way round that in patients who are ill already, no extra time should be spent on risk during PCI). That would be of my interest anyway. 

4. Table 2. just to be sure: in METHODS, the MACE endpoint is (line 95): mortality, nonfatal MI, nonfatal stroke, acute heart failure, ischemia driven revascularisation, major bleeding and acute renal failure. In Table 2 however, MACE is defined as 30 day CV mortality. So what was it: all-cause mortality or only CV mortality. If all cause, please omit CV in Table 2. 

5. Discussion: Line 215-239. You discuss all the studies that you mention in introduction, but you do not compare it to your own study. Maybe you better shorten this section to the relevant comparisons with your own study. What I found relevant is the line 228-229, in which the PRAMI trial apparently did have some succes with an immediate complete revascularisation, compared to culprit-lesion only. Also, REF 7 adds on timing of non-culprit lesion revascularization, in which no difference in outcome was found between the two approaches. This is relevant, not ? Your own study seems to indicate a difference with these studies, not showing an advantage for immediate complete revascularisation, what is the difference between your study and the study with ref 7. Your discussion starts at line 261, this can be done earlier; and after line 261you now refer to the 4 meta-analyses which comply more with your results. Maybe you start the discussion what is really known for the question that you wanted to answer: immediate or staged intervention for patients, and if the PRAMI and REF 7 are included in the metaanalyses (which of course tell the same things 4 times), you can have a shorter (and more interesting) discussion.  

6. text

Line 41, references are specific to the sentences: after "is well established" (REF 3-5), and line 42 after "intervention for these patients" (REF 6-9) and line 43 guidelines ... intervention" (REF 10-12). 

Line 42, "clear benefit of interventions for these patients" should be clear benefit of multivessel interventions for these patients.

Line 45 contradictive: probably meant is "is a matter of debate" . 

Line 46-49: it is not so obvious that prognosis was dependent on Syntax score until the Syntax score was developed. So maybe change this into: The syntax score was developed that could predict prognosis... 

Line 82: This study confirms the principles of Helsinki declaration: probably meant is "conforms to the principles of the Helsinki declaration" .

Line 95: were identified as:  omit identified and put a : before all cause mortality. 

Line 107: add reference to RIFLE definition of kidney injury. 

Line 163 in Table 1: compaired = compared. 

Line 191 Figure 2, contrubuted = contributed.

Line 211-214: you added a sentence on predictors of MACE. The last sentence "Our restrospective small study may be insufficient to demonstrate these interactions" : you are not discussing the interactions in this sentence but the associations. It is contradictive to your conclusions, and I would suggest to omit this last sentence "Our retrospective small study ..interactions". In the limitation section, you already mention the retrospective nature of the study.

7. Title: you may consider changing the title to "Comparison of 30 day MACE between immediate versus staged complete revascularization in acute myocardial infarction with mutlivessel disease and the effect of coronary lesion complexity" 

Author Response

Dear Rewiever

Thank you for your countributions and comments to our manuscript. In the manuscript we made the necessary changes in line with your suggestions. Corrected manuscript was directed to you.

Your sincerly

1.       Patient selection:

We made the changes in unsuccessful PCI and in-hospital MACE that we have excluded in the study in the way you want, in Figure 1 and text. Related  numbers were added to the text. In the introduction and methodology we mentioned that the study included successful multi-vessel intervention patients. We have added that the exclusion of unsuccessful PCI and in-hospital MACEs in the Limitations section is a limitation of our study.

2.         MACE at 30 days definition:

We described our clinical outcomes as MACE within 30 days after discharge. We have made the necessary changes in the abstract, conclusion and other places you specify.

We didn't add these numbers to the results again because we gave the number of in-hospital MACE  in the exclusion criterias in the methodology section.

in-hospital deaths were added in figure 1 .

3.       In the Methods:

The corrections you specified were made in the corresponding rows. In our study, the decision and timing of nonculprit vessel intervention were determined by the operators. As you mentioned, patient and clinic (LVEF etc) were effective in these preferences.

4.       Table 2:

We evaluated cardiovascular mortality as mortality. we added cardiovascular to text line 95.

5.       Discussion:

In the discussion, the places that were considered unnecessary were removed, the differences of our study were emphasized and some more fluent as you mentioned.

6.       Text

We changed the places of references in line 41 and 42

We added ‘multivessel’ to line 42

We changed contradictive  to  ‘a matter of debate’

Changes have been made about  Sx score on Line 46-49

We changed ‘confirms’ to ‘conforms in line 82

We delete ‘identified as’ at line 95

We added reference related to RIFLE  (sixteenth reference). Number of other references changed.

We changed ‘compaired’ to ‘compared’ in table 1

We changed ‘contrubuted’ to ‘contributed’ in figure 2

We deleted the sentences at line 211-214 as you wanted

We cahnged the title as you comments